# Non-Equilibrium Enhancement of Classical Information Transmission

**DOI:** 10.3390/e26070581

**Published:** 2024-07-08

**Authors:** Qian Zeng, Jin Wang

**Affiliations:** 1State Key Laboratory of Electroanalytical Chemistry, Changchun Institute of Applied Chemistry, Changchun 130022, China; qzeng@ciac.ac.cn; 2Department of Chemistry and Physics, State University of New York, Stony Brook, NY 11794, USA

**Keywords:** information transmission, mutual information, information channel capacity, nonequilibrium information driving force, entropy production

## Abstract

Information transmission plays a crucial role across various fields, including physics, engineering, biology, and society. The efficiency of this transmission is quantified by mutual information and its associated information capacity. While studies in closed systems have yielded significant progress, understanding the impact of non-equilibrium effects on open systems remains a challenge. These effects, characterized by the exchange of energy, information, and materials with the external environment, can influence both mutual information and information capacity. Here, we delve into this challenge by exploring non-equilibrium effects using the memoryless channel model, a cornerstone of information channel coding theories and methodology development. Our findings reveal that mutual information exhibits a convex relationship with non-equilibriumness, quantified by the non-equilibrium strength in transmission probabilities. Notably, channel information capacity is enhanced by non-equilibrium effects. Furthermore, we demonstrate that non-equilibrium thermodynamic cost, characterized by the entropy production rate, can actually improve both mutual information and information channel capacity, leading to a boost in overall information transmission efficiency. Our numerical results support our conclusions.

## 1. Introduction

Information transmission is critical in a physical/biological world and in human society. Examples include communication [1,2,3], cryptography [4,5,6], information transmission in cells [7,8,9], information processing in the brain [10,11,12,13], and information propagation in social media and networks [14,15]. Recently, quantum information transmission has received much attention for the new generation of information communications and design for quantum computers [3]. The efficiency of information transmission is often measured by the mutual information between two physical systems. Information capacity is defined as the upper bound of mutual information. The study of information transmission has focused on various closed systems. However, most of the practical examples are open systems. Despite its huge successes in closed systems, there are still challenges for open systems on how non-equilibrium effects characterized by the exchange of energy, information, and materials from outside environments influence mutual information and information capacity and, therefore, information transmission efficiency [16].

The memoryless channel is the simplest information transmission model in information and communication theory [17]. It depicts how a noisy channel distorts the transmitted symbols with a transmission (conditional) probability distribution, which is permuted according to the transmitted symbol. This model is also important because it is the cornerstone of the information channel coding theories and associated methodology development that are concerned with how to transmit bit information more efficiently. It is noteworthy that physically noisy channels often lead to information dissipation or information loss of the transmitted messages.

A memoryless channel can be viewed as an ensemble of non-equilibrium open systems. This is because the information transmission can be through different noisy channels or the same channel with difference transmissions on the signal components. By quantifying the non-equilibrium strength [18,19] for information dynamics and non-equilibrium entropy production [20,21,22] or the dissipation cost for information thermodynamics [23,24,25,26] we investigate the non-equilibrium dynamics and the thermodynamics of this class of channels. The channel system is shown to be at equilibrium state if and only if the non-equilibrium strength, in terms of transmission probabilities and the corresponding entropy production rate, is zero. Otherwise, the system is in a non-equilibrium state.

Our investigation unveils a connection between the model’s mutual information, non-equilibrium strength, and entropy production. We reveal that mutual information exhibits a convex relationship with non-equilibriumness, quantified by non-equilibrium strength via probability flux. Notably, channel information capacity, which reflects the maximum achievable information transfer, is also enhanced by non-equilibrium effects. Furthermore, information cost, captured by the entropy production rate, surprisingly boosts both mutual information and information capacity, leading to a net gain in information transmission efficiency. This implies that non-equilibrium strength, mediated by probability flux, and the information-theoretic cost of non-equilibrium thermodynamics can both improve communication efficiency. However, at equilibrium, the absence of non-equilibrium effects results in vanishing mutual information and information capacity, signifying no effective information transmission. Our numerical simulations corroborate these findings, validating the model’s key features.

## 2. Non-Equilibrium Information Dynamics for Memoryless Channel

### 2.1. Information Dynamics for Memoryless Channel

Let us suppose that an information sender sends a message randomly to an information receiver within time *t*. The message is denoted by $S=\{s_{1},s_{2},...,s_{t}\}$, where $s_{i}$ is the input symbol chosen from the code book S. Here, we assume that the input symbols *s* in one message are sent independently according to an identical distribution P(s), which is the so-called input distribution. During the transmission, the message *S* is usually interfered with by random environments, it is received as a noisy message $X=\{x_{1},x_{2},...,x_{t}\}$ (where $x_{i}$ is the noisy output symbol), and it forms a noisy code book X, which must not be the same as the original code book S. The random environments are defined as the communication channel in the information theory [27,28]. The influence of the channel on the messages is characterized by the conditional distribution P(X|S). The channel is said to be memoryless if it distorts each symbol independently of other symbols. Then, we can quantify the channel by using the time-invariant (conditional) transmission probability distribution q(x|s) for the output and input symbols *x* and *s* (see Figure 1). Thus, the conditional probability distribution P(X|S) for the messages can be rewritten into the products of the conditional transmission probabilities $q(x_{i}|s_{i})$ for the symbols at each time, $P\left(X|S\right)=\prod_{i=1}^{t}q\left(x_{i}|s_{i}\right)$. The probability of the output symbol *x* can be given by the input distribution and the transmission distribution: (1)$$\begin{array}{c} P\left(x\right)=\sum_{s}q\left(x|s\right)P\left(s\right). \end{array}$$

The P(x) is the so-called output distribution. In this model, we assume that both the sent and received messages *S* and *X* are stationary, independently and identically distributed processes with stationary distributions of symbols P(s) and P(x), respectively.

From another point of view, information transmission can be viewed as a process for information processing (see Figure 2a). Here, the channel can be regarded as an information processor controlled by random environments. When the sender inputs a symbol *s* into the channel at time *t*, the channel processes the symbol randomly for a period or delay δt=1 and then outputs a new symbol *x* to the receiver at time t+1. With this perspective, we can establish Markovian information dynamics for the memoryless channel model. The symbols $s_{t}$ and $x_{t}$ can be viewed as the information state of the composite system of the sender and the channel, $z_{t}=\left(s_{t},x_{t}\right)$. The output symbol $x_{t}$ is decided by the channel within the information processing, according to the transmission probability $q(x_{t}|s_{t-1})$. The input symbol $s_{t}$ is independent of $s_{t-1}$ and is generated by the sender according to the probability $P(s_{t})$. In addition, both $x_{t}$ and $s_{t}$ are independent of $x_{t-1}$ if the channel is memoryless. Then, the transition probability of the state $z_{t}$ given $z_{t-1}$ can be written as follows:(2)$$\begin{array}{c} Q\left(z_{t}|z_{t-1}\right)\equiv Q\left(s_{t},x_{t}|s_{t-1}\right)=P\left(s_{t}\right)q\left(x_{t}|s_{t-1}\right). \end{array}$$With this transition probability, the time evolution of the state distribution P(z) can be given by the following Markov master equation:(3)$$\begin{array}{c} P\left(z_{t}\right)=\sum_{z_{t-1}}Q\left(z_{t}|z_{t-1}\right)P\left(z_{t-1}\right). \end{array}$$

Here, the transition probability *Q* in Equation (Equation 2) is recognized as the information driving force behind the composite system [18,19]. The driving force *Q* describes how the channel processes an input symbol $s_{t-1}$ and generates an output $x_{t}$; *Q* also determines the evolution of the probability distribution P(z) in the dynamics. It can be verified that given an arbitrary initial distribution $P(z_{1})$ the composite system finally goes to the unique stationary joint distribution π(z)=P(s)P(x), which is independent of time, such that $\pi \left(z\right)=\sum_{z}Q\left(z|z^{\prime }\right)\pi \left(z^{\prime }\right)$.

Although the information driving force *Q* is associated with the input distribution P(s), the efficiency of the information transfer is more related to the transmission probability of the channel, q(x|s). Thus, we focus our discussions and conclusions mainly on how q(x|s) influences information transfer with an arbitrary given input P(s).

### 2.2. Non-Equilibrium Strength

The environments of the channel are usually complex, which can give rise to non-equilibriumness in the channel, reflected in the transmission probabilities q(x|s). This is equivalent to the measure of the broken time reversal symmetry or time-irreversibility in the composite dynamics. That is to say, the probability of the time sequence of the information states $Z=\{z_{1},z_{2},...,z_{t}\}$, as shown in Equation (Equation 2), is usually not equal to that of the corresponding time reversal sequence $\tilde{Z}=\left\{z_{t},z_{t-1},...,z_{1}\right\}$. It is noteworthy that according to the information dynamics in Equation (Equation 3) each output symbol $x_{i}$ is determined by the input symbol $s_{i-1}$ via the transmission probability $q(x_{i}|s_{i-1})$ in the forward time sequence *Z*; and $x_{i}$ is decided by the input $s_{i+1}$, according to the transmission probability $q(x_{i}|s_{i+1})$ in the backward time sequence $\tilde{Z}$ (see Figure 2b). Then, the time-irreversibility of these information dynamics depends on the difference between the probability of the transition from $s_{i-1}$ to $x_{i}$ in the forward time direction and that of the transition from $s_{i+1}$ to $x_{i}$ in the backward time direction. Furthermore, the output $x_{i}$ is independent of the previous output $x_{i-1}$ in the time forward transition and the time successive output $x_{i+1}$ in the time backward transition; also, $x_{i}$ is independent of the simultaneous input $s_{i}$, due to the information dynamics in Equation (Equation 3). Thus, the influences of the outputs $x_{i-1}$ and $x_{i+1}$, including the input $s_{i}$, on the time-irreversibility can be negligible. The above observation yields the quantification of the time-irreversibility represented by the probability flux, which can be given by the difference between the probability of a time sequence and that of the time-reversal time sequence in the following equation:(4)$$\begin{array}{c} J_{x_{i}}\left(s_{i-1},s_{i+1}\right)=\sum_{x_{i-1},x_{i+1},s_{i}}\left[P\left(Z\right)-P\left(\tilde{Z}\right)\right]=2P\left(s_{i-1}\right)P\left(s_{i+1}\right)d_{x_{i}}\left(s_{i-1},s_{i+1}\right), \end{array}$$
with
(5)$$\begin{array}{c} d_{x_{i}}\left(s_{i-1},s_{i+1}\right)=\frac{1}{2}\left[q\left(x_{i}|s_{i-1}\right)-q\left(x_{i}|s_{i+1}\right)\right]. \end{array}$$

Here, $Z=\left\{z_{i-1},z_{i},z_{i+1}\right\}=\left\{\left(s_{i-1},x_{i-1}\right),\left(s_{i},x_{i}\right),\left(s_{i+1},x_{i+1}\right)\right\}$ is a time sequence that contains the transition from $s_{i-1}$ to $x_{i}$ in the forward time direction. The final state $z_{i+1}$ in the sequence *Z* is needed in the quantification of the time-irreversibility, because by using *Z* the transition form $s_{i+1}$ to $x_{i}$ in the backward time direction can be contained in the corresponding time-reversal time sequence $\tilde{Z}=\left\{\left(s_{i+1},x_{i+1}\right),\left(s_{i},x_{i}\right),\left(s_{i-1},x_{i-1}\right)\right\}$. Then, the probabilities of *Z* and $\tilde{Z}$, which can be given by Equation (Equation 2), guarantee the dynamical connection between the transitions from $s_{i-1}$ to $x_{i}$ and form $s_{i+1}$ to $x_{i}$, which is decided by the information dynamics in Equation (Equation 3). Furthermore, the input symbols $x_{i-1}$ and $x_{i+1}$ and the output symbol $s_{i}$ are summed away from the probabilities of *Z* and $\tilde{Z}$ because they are not important to the time-irreversibility, as discussed in the above.

The probability flux *J* can be used to characterize the time-irreversibility of the composite information dynamics in Equation (Equation 3) because of term *d* in Equation (Equation 5), which reflects the non-equilibrium strength of the information dynamics (detailed discussion can be found in Appendix A). The dynamics reach equilibrium if and only if the probability flux *J* or the non-equilibrium strength *d* vanishes; otherwise, the dynamics are in non-equilibrium.

### 2.3. Non-Equilibrium Decomposition of Transmission Probabilities

From the perspective of non-equilibrium information dynamics, the non-equilibrium strength *d* in Equation (Equation 5) suggests a decomposition of the transmission probabilities q(x|s), as follows:(6)$$\begin{array}{c} q\left(x|s\right)=m_{x}\left(s,s^{\prime }\right)+d_{x}\left(s,s^{\prime }\right),\;\mathrm{for}\;s^{\prime }\neq s. \end{array}$$
where *m* and *d* can be given in the consistent forms in the force decomposition shown in Appendix A:(7)$$\begin{array}{c} \left\{\begin{array}{c} m_{x}\left(s,s^{\prime }\right)=\frac{1}{2}\left[q\left(x|s\right)+q\left(x|s^{\prime }\right)\right],\\ d_{x}\left(s,s^{\prime }\right)=\frac{1}{2}\left[q\left(x|s\right)-q\left(x|s^{\prime }\right)\right], \end{array}\right. \end{array}$$Here, *m* is recognized as the equilibrium strength of the information dynamics, in contrast to the effect of *d*.

In Equation (Equation 6), $s^{\prime }$ represents an input symbol which differs from the input symbol *s*. Due to this decomposition, both *m* and *d* can change independently of each other within the ranges of constraints that are properly given. Consequentially, q(x|s) can be changed by changing *m* and *d*. This can improve the information transfer as discussed in the above.

Due to the non-negativity and normalization of the transmission probabilities q(x|s), the constraints can be given as follows:(8)$$\begin{array}{c} \left\{\begin{array}{c} 0\leq m_{x}\left(s,s^{\prime }\right)+d_{x}\left(s,s^{\prime }\right)\leq 1\\ 0\leq m_{x}\left(s,s^{\prime }\right)\leq 1\\ \sum_{x}m_{x}\left(s,s^{\prime }\right)+d_{x}\left(s,s^{\prime }\right)=1 \end{array}\right., \end{array}$$The first and second constraints are originated from the non-negativity of q(x|s); that is: $0\leq q\left(x|s\right)=m_{x}\left(s,s^{\prime }\right)+d_{x}\left(s,s^{\prime }\right)\leq 1$ and $0\leq m_{x}\left(s,s^{\prime }\right)=\frac{1}{2}\left[q\left(x|s\right)+q\left(x|s^{\prime }\right)\right]\leq 1$. The third constraint comes from the normalization of q(x|s), i.e., $1=\sum_{x}q\left(x|s\right)=\sum_{x}m_{x}\left(s,s^{\prime }\right)+d_{x}\left(s,s^{\prime }\right)$. It is noteworthy that for an appropriate set of the decomposition that satisfies the constraints in Equation (Equation 8), all the possible non-equilibrium strengths $d_{x}\left(s,s^{\prime }\right)$ form a convex set by fixing the equilibrium strength $0\leq m_{x}\left(s,s^{\prime }\right)\geq 1$ at each *x* (see Appendix B).

A strong connection exists between non-equilibrium strength and information transfer. Intuitively, when composite dynamics reach equilibrium (detailed discussion in Appendix A) the non-equilibrium strength *d* becomes zero. This signifies that the channel generates output symbols (*x*) independent of the sender’s input. In other words, the transmission probabilities q(x|s) become equivalent to the output distribution P(x), regardless of the input distribution P(s). This implies that under equilibrium, no matter the input distribution, the channel transmits no useful information from the sender. Conversely, a non-equilibrium state exists only when the non-equilibrium strength *d* is not zero. This can lead to a scenario where q(x|s) deviates from P(x) for any input distribution. Consequently, information transfer can only occur under non-equilibrium conditions.

## 3. Information Transmission Enhanced by Non-Equilibrium Strength

### 3.1. Mutual Information and Non-Equilibrium Strength

Information theory gives a fundamental quantification of the useful information transferred from the information sender to the receiver through the memoryless channel. This is the so-called information transfer rate or mutual information rate [29,30] per symbol for the two messages $X=\{x_{1},x_{2},...,x_{t}\}$ and $S=\{s_{1},s_{2},...,s_{t}\}$:(9)$$\begin{array}{ccc} I & = & \lim_{t\to \infty }\frac{1}{t}\sum_{S,X}P\left(S,X\right)\log\frac{P(S,X)}{P(S)P(X)}\\ & = & \sum_{z^{\prime },z}\pi \left(z^{\prime }\right)Q\left(z|z^{\prime }\right)\log\frac{\pi \left(z^{\prime }\right)Q\left(z|z^{\prime }\right)}{\pi \left(z^{\prime }\right)\pi \left(z\right)}\\ & = & \sum_{x,s}P\left(s\right)q\left(x|s\right)\log\frac{q(x|s)}{P(x)}\geq 0. \end{array}$$

In the second equality in Equation (Equation 9), z=(x,s) and $z^{\prime }=\left(x^{\prime },s^{\prime }\right)$; here, *I* is written in terms of the Markovian information dynamics in Equation (Equation 3), where the channel can be viewed as an information processing device and the stationary distribution π(s,x)=P(s)P(x). In the third equality, *I* is in the ordinary form of the mutual information between the input and output symbols. It can be proved that these two expressions of the mutual information rate are equivalent to each other from the descriptions in Equations (2)–(4). Here, I>0 means that although the channel is noisy the useful information from the sender can be received by the receiver. Otherwise, the vanishing *I* indicates that the useful information is totally corrupted through the channel. Here, the mutual information achieves its minimum (I=0) if and only if the input and output symbols are independent of each other, i.e., q(x|s)=P(x) holds for every *x* and *s*. This observation is consistent with the perspective of non-equilibrium information dynamics: ineffective information transfer (I=0) implies a detailed balance condition in the channel, where the non-equilibrium strength d=0 in this situation. On the other hand, when in equilibrium the information cannot be transferred efficiently. In other words, the information from the sender can be received by the receiver if and only if the channel model is in non-equilibrium. This inspires us to investigate the relationship between non-equilibriumness and information transfer.

We catch the first glimpse of this relationship by noting that the mutual information *I* in Equation (Equation 9) can be rewritten as the function with respect to the non-equilibrium strength *d* with the decomposition in Equation (Equation 6):(10)$$\begin{array}{ccc} I(d) & = & \sum_{x,s}P\left(s\right)q\left(x|s\right)\log\frac{q(x|s)}{P(x)}\\ & = & \sum_{x,s}P\left(s\right)\left[m_{x}\left(s,s^{\prime }\right)+d_{x}\left(s,s^{\prime }\right)\right]\log\frac{\left[m_{x}\left(s,s^{\prime }\right)+d_{x}\left(s,s^{\prime }\right)\right]}{\sum_{s}P\left(s\right)\left[m_{x}\left(s,s^{\prime }\right)+d_{x}\left(s,s^{\prime }\right)\right]} \end{array}$$

Here, $P\left(x\right)=\sum_{s}P\left(s\right)q\left(x|s\right)=\sum_{s}P\left(s\right)\left[m_{x}\left(s,s^{\prime }\right)+d_{x}\left(s,s^{\prime }\right)\right]$ shown in Equation (Equation 1). According to Equation (Equation 31), the model is in equilibrium if and only if every strength d=0, and this is also the necessary and sufficient condition of ineffective information transfer (I=0), no matter how the input P(s) is chosen. Thus, with the arbitrarily chosen P(s), we can describe the impact of the channel-induced non-equilibriumness, characterized by the strength *d*, on the information transfer, as follows.

By noting the constraints on *m* and *d* in Equation (Equation 8), we can conclude that *I* is a convex function of the non-equilibrium strength *d* at each fixed equilibrium strength *m* and input P(s) (see Appendix C). The convexity of I(d) indicates that *I* has the unique minimum I=0 at d=0, where the channel model is under equilibrium; in addition, *I* is a monotonously increasing function as the absolute value of $|d_{x}\left(s,s^{\prime }\right)|$ increases. This uncovers the intrinsic relationship between the information transfer and the non-equilibriumness: the degree of the non-equilibriumness *d* determined by the channel enhances the information transfer *I*. This conclusion can be independent of the input distribution P(s).

### 3.2. Channel Capacity and Non-Equilibrium Strength

The channel coding theorem [26,27] reveals that the information transfer rate cannot exceed an upper bound, the channel capacity, which is exactly the maximum of the mutual information *I*, and merely depends on the channel. This yields a unique optimal input distribution $P^{*}\left(s\right)$ at which the capacity can be achieved. On the other hand, as with the relationship between the mutual information and non-equilibriumness revealed in Equation (Equation 7), we are also interested how the non-equilibriumness influences the capacity. Mathematically, the information capacity is formulated as the maximization of the mutual information *I*:$$\begin{array}{ccc} C & = & \max_{P(s)}I;\\ \mathrm{subject}\;\mathrm{to}\; & & \sum_{s}P\left(s\right)=1,\;\mathrm{and}\;P\left(s\right)\geq 0. \end{array}$$

Since the information capacity *C* is a concave function of P(s) then it has an unique maximum with respect to an unique optimal input $P^{*}\left(s\right)$. Although this optimization problem, in general, does not have an analytical solution [31] we can investigate the relationship between *C* and the non-equilibrium strength *d*, which characterizes the channel-induced non-equilibriumness in the analytically solvable cases or we can perform a numerical test for more general cases. For example, the (weak) symmetric channel model is solvable in the capacity problem, and it is usually used as an explanatory case in the information theory. In the symmetric channel model, the transmission distribution q(x|s) at each input *s* is a permutation of that at the other input *s*. Thus, the information capacity can be achieved at the uniform distribution of *s*, $P^{*}\left(s\right)=1/N_{s}$ ($N_{s}$ is the number of the symbols *s*). The information capacity can then be given as follows:(11)$$\begin{array}{c} C=\log N_{x}-S, \end{array}$$
where $N_{x}$ is the number of the symbol *x*, and where $S=-\sum_{x}q\left(x|s\right)\log q\left(x|s\right)$ is the entropy of the arbitrary transmission distribution. Due to the symmetry in the channel, all the entropies *S* are equal to each other. To see the relationship between the capacity and the non-equilibrium strength *d* in this model, we re-express *C* as the function of *d*:(12)$$\begin{array}{ccc} C & = & \log N_{x}+\sum_{x}\left[m_{x}\left(s,s^{\prime }\right)+d_{x}\left(s,s^{\prime }\right)\right]\log\left[m_{x}\left(s,s^{\prime }\right)+d_{x}\left(s,s^{\prime }\right)\right], \end{array}$$
where we decompose q(x|s) by using Equation (Equation 6). Then, by fixing the equilibrium strength *m* in Equation (Equation 12) the information capacity *C* is a convex function of *d*. Meanwhile, the transmission probabilities are $q\left(x|s\right)=P\left(x\right)=\frac{1}{N_{x}}$, and there is no useful information that can be transferred by the channel. Otherwise, the capacity can be enhanced (monotonously increased) by the increasing absolute values of $|d_{x}\left(s,s^{\prime }\right)|$ in the symmetric model.

## 4. Better Information Transfer Efficiency under Larger Dissipation

### 4.1. Information Dissipation Enlarges Mutual Information

The larger strength of the non-equilibrium force *d*, which characterizes a non-equilibrium information dynamic (Equation (Equation 3)), can increase the efficiency or the rate of the transferred information *I* in the channel. On the other hand, a non-equilibrium information dynamic usually gives rise to information dissipation. Here, we are interested in how information dissipation influences information transfer.

In thermodynamics, the rate of thermodynamic dissipation is characterized by the entropy production rate (EPR). In this model, EPR can be given by the averaged log ratio of the probabilities of the time sequences and that of the time-reversal sequences and by taking the time average as follows, according to the fluctuation theorem [21]:(13)$$\begin{array}{c} R_{Z}=\lim_{t\to \infty }\frac{1}{t}\sum_{Z}P\left(Z\right)\log\frac{P(Z)}{P(\tilde{Z})}, \end{array}$$
where $Z=\left\{z_{1},z_{2},...,z_{t}\right\}=\left\{\left(s_{1},x_{1}\right),\left(s_{2},x_{2}\right),...,\left(s_{t},x_{t}\right)\right\}$ is a time sequence of the composite dynamics described by Equation (Equation 3) and $\tilde{Z})$ is the time-reversal sequence of *Z*. Since the information dynamics are assumed to be Markovian, the expression of the EPR can be evaluated by using the probability of the time sequences shown in Equation (Equation 2):(14)$$\begin{array}{ccc} R_{Z} & = & \frac{1}{2}\sum_{s_{i-1},s_{i+1},x_{i}}J_{x_{i}}\left(s_{i-1},s_{i+1}\right)\log\frac{Q(s_{i},x_{i}|s_{i-1})}{Q(s_{i},x_{i}|s_{i+1})}\\ & = & \sum_{s,s^{\prime },x}P\left(s\right)P\left(s^{\prime }\right)q\left(x|s\right)\log\frac{q(x|s)}{q(x|s^{\prime })}\geq 0. \end{array}$$

Here, the first equality in Equation (Equation 14) is written from the perspective of non-equilibrium Markov information dynamics, where the time-irreversible probability flux $J_{x_{i}}\left(s_{i-1},s_{i+1}\right)$ is given by Equation (Equation 4) and the transition probability *Q* is given by Equation (Equation 2). From this perspective, the EPR is quantified by the averaged information difference or information loss between the transition in the backward time direction and the transition in the forward time direction, given by $-\log Q(s_{i},x_{i}|s_{i+1})$ and $-\log Q(s_{i},x_{i}|s_{i-1})$, respectively. The information loss $l=\log\frac{Q(s_{i},x_{i}|s_{i-1})}{Q(s_{i},x_{i}|s_{i+1})}$ can be positive if the forward transmission probability is larger than the backward probability, $Q\left(s_{i},x_{i}|s_{i-1}\right)>Q\left(s_{i},x_{i}|s_{i+1}\right)$. This means that the forward transition happening is more likely than the backward transition in time with a positive information loss. The information loss l<0 or, equivalently, $Q\left(s_{i},x_{i}|s_{i-1}\right)<Q\left(s_{i},x_{i}|s_{i+1}\right)$ indicates that the forward transition happening is less likely than the backward transition in time with a negative information loss. Otherwise, l=0 if and only if $q\left(x_{i}|s_{i-1}\right)=q\left(x_{i}|s_{i+1}\right)$; then, the forward and backward transition happens with the same probability and with no information loss. Physically, the EPR can be regarded as the total entropy change consisting of the system and the environment [22]. Due to the Second thermal law, the total entropy change or the EPR cannot be negative, while some individual information losses *l* in the EPR can be negative. In other words, individual losses can be negative while the average losses must be non-negative. Mathematically, the non-negativity of the EPR is guaranteed by the form of the relative entropy $D_{KL}(P\left(Z\right)||P\left(\tilde{Z}\right))\geq 0$ in Equation (Equation 13). The non-negativity indicates that, driven by the non-vanishing probability flux *J*, the EPR is always positive in non-equilibrium conditions. Otherwise, the EPR vanishes at the equilibrium, and the probability flux *J* vanishes. Thus, the non-equilibriumness or time-irreversibility of the information dynamics can lead to positive information loss or thermodynamic dissipation quantified by the EPR.

The second equality in Equation (Equation 14) is written from the perspective of the information transfer. In this perspective, the information *i* of the input symbol can reduce the uncertainty of the output symbol; thus, the information *i* of an input symbol *s* carried by an output symbol *x* can be given by the reduction of the uncertainty of *x*, $i_{s}=\log\frac{q(x|s)}{P(x)}$. Here, −logP(x) quantifies the prior uncertainty of *x* without the information of *s*, and −logq(x|s) quantifies the posterior uncertainty of *x*, which has been reduced by the information $i_{s}$. Then, the entity $l=\log\frac{q(x|s)}{q(x|s^{\prime })}=\log\frac{q(x|s)}{P(x)}-\log\frac{q(x|s^{\prime })}{P(x)}=i_{s}-i_{s^{\prime }}$ in Equation (Equation 14) quantifies the difference between the efficient information $i_{s}$ of an input *s* carried by the output $x^{\prime }$ and the efficient information $i_{s^{\prime }}$ of another input $s^{\prime }\neq s$ contained in *x*. If the output *x* carries more information of *s* than that of $s^{\prime }$, then $l=i_{s}-i_{s^{\prime }}>0$. On the contrary, l<0 indicates that *x* carries more information of $s^{\prime }$ than that of *s*. Either the positive or negative information difference *l* implies that if the output *x* carries more information of one input *s* than that of another input $s^{\prime }$ then *s* can be recovered at the receiver more efficiently than $s^{\prime }$. This indicates that the positions of *s* and $s^{\prime }$ in the (second equality) EPR are symmetric. Then, a negative information difference *l* is equivalent to a positive *l* by exchanging the positions of *s* and $s^{\prime }$. Thus, the averaged information difference or the EPR must be non-negative. The form of the relative entropy $D_{KL}(P\left(Z\right)||P\left(\tilde{Z}\right))\geq 0$ in Equation (Equation 13) also guarantees the non-negativity of the EPR. Otherwise, l=0 refers to the worst situation, where the *x* carries equivalent information of both $s^{\prime }$ and *s*. Thus, $s^{\prime }$ and *s* cannot be distinguished by the receiver when the output *x* is observed. The worst situation can happen when the information dynamics are in equilibrium, i.e., the detailed balance condition in Equation (Equation 31) is achieved. Furthermore, the information dissipation characterized by the EPR vanishes at equilibrium. Otherwise, the useful information of the input symbols can be transferred by the output symbols only in non-equilibrium conditions, which is characterized by the positive EPR in the thermodynamics. Thus, the EPR in Equation (Equation 14) uncovers the intrinsic connection between the non-equilibrium information thermodynamics and the information transfer.

To see the relationship between the EPR and the information transfer more clearly, we note that the information transfer rate in Equation (Equation 9) can be decomposed into a time-reversible part $I_{M}$ and a time-irreversible part $I_{D}$ with the decomposition in Equation (Equation 6), as follows:(15)$$\begin{array}{c} I=I_{M}+I_{D} \end{array}$$
with
(16)$$\begin{array}{c} \left\{\begin{array}{c} I_{M}=\sum_{x,s}P\left(s\right)m_{x}\left(s,s^{\prime }\right)\log\frac{\left[m_{x}\left(s,s^{\prime }\right)+d_{x}\left(s,s^{\prime }\right)\right]}{\sum_{s}P\left(s\right)\left[m_{x}\left(s,s^{\prime }\right)+d_{x}\left(s,s^{\prime }\right)\right]}\\ I_{D}=\sum_{x,s}P\left(s\right)d_{x}\left(s,s^{\prime }\right)\log\frac{\left[m_{x}\left(s,s^{\prime }\right)+d_{x}\left(s,s^{\prime }\right)\right]}{\sum_{s}P\left(s\right)\left[m_{x}\left(s,s^{\prime }\right)+d_{x}\left(s,s^{\prime }\right)\right]}. \end{array}\right. \end{array}$$

Here, $I_{M}$ and $I_{D}$ correspond to the equilibrium strength *m* and the non-equilibrium strength *d*, respectively. The direct connection between the information transfer rate and the EPR can be uncovered via the time-irreversible mutual information rate $I_{D}$ by noting the identity [18,19]:$$\begin{array}{c} R_{Z}=R_{X}+R_{S}+2I_{D}. \end{array}$$

Here, $R_{X}$ and $R_{S}$ are the EPR of the information sender and the receiver, respectively, and they can be negligible if both the sent messages and received messages are independently and identically distributed processes. Thus, the information dissipation in the information processing can be exactly quantified by the time-irreversible mutual information rate $I_{D}$:(17)$$\begin{array}{c} I_{D}=\frac{1}{2}R_{Z}. \end{array}$$
where the factor 1/2 is due to the symmetry between the forward and backward time sequences in the expression of the EPR in Equation (Equation 13). By noting the relation in Equation (Equation 17), we obtain
(18)$$\begin{array}{c} I=I_{M}+I_{D}=I_{M}+\frac{1}{2}R_{Z}. \end{array}$$

This equation implies that the efficiency of the information transfer can be enhanced by increasing the information dissipation. As shown in Equation (Equation 15), the time-reversible mutual information rate $I_{M}$ and the time-irreversible mutual information rate $I_{D}$ are in the form of the functions of the degree of the non-equilibriumness *d*. By fixing *m* and the input P(s), $I_{D}$ is a convex function of *d*, and it monotonously increases as the absolute value $|d_{x}\left(s,s^{\prime }\right)|$ increases. The convexity of $I_{D}$ can be verified by the convexity of the information transfer rate *I*, where the proof has been given in Appendix C.

By noting the same nature of the mutual information rate *I* as the function of *d* in Equation (Equation 10), we conclude that an increase in the efficiency of the information transfer requires an increase in the information dissipation cost. Thus, thermodynamic cost can be used to boost information transfer efficiency.

### 4.2. Better Information Capacity under Larger Dissipation

As shown in Equation (Equation 12), information capacity can be enhanced by the non-equilibriumness of the symmetric model. Also, it can be seen that the EPR or the information dissipation at the optimal input $P^{*}\left(s\right)=\frac{1}{N_{s}}$ is a convex function of the degree non-equilibriumness of the model. By inserting $P^{*}\left(s\right)$ into Equation (Equation 14) or Equation (Equation 15), we have the following:(19)$$\begin{array}{c} I_{D}^{*}\left(d\right)=\sum_{x}d_{x}\left(s,s^{\prime }\right)\log\left[m_{x}\left(s,s^{\prime }\right)+d_{x}\left(s,s^{\prime }\right)\right]. \end{array}$$

By fixing the averaged equilibrium strength *m*, the convexity of $I_{D}^{*}$ with respect to the averaged non-equilibrium strength *d* can be verified by noting the convexity of the information capacity clearly.

Through observation of the information capacity expression in Equation (Equation 12) and the EPR expression in Equation (Equation 14), one can obtain the relationship between the information capacity and the EPR:(20)$$\begin{array}{c} C=I_{M}^{*}+I_{D}^{*}. \end{array}$$
where $I_{M}^{*}=\log N_{x}+\sum_{x}m_{x}\left(s,s^{\prime }\right)\log\left[m_{x}\left(s,s^{\prime }\right)+d_{x}\left(s,s^{\prime }\right)\right]$ is the time-reversible part in the information capacity. We can see that in the symmetrical channel the information capacity increases with respect to the increase of the thermodynamic information dissipation cost. Therefore, the thermodynamic cost can be used to increase the information capacity.

## 5. A Case with a Binary Memoryless Channel

To illustrate our concept, let us consider a simple model with a binary information measurement, which can be viewed as a fundamental form of information transmission. All the elements are explicitly defined and readily interpretable. In this model, the object, acting as the information sender, can exist in two states: s=0 or s=1, which can be viewed as the input symbols. These states (symbols) influence the measurement device, acting as the information receiver, through a potential field within a confined area. This potential landscape has two “flat” wells: a higher one with height H>0 and a lower one with height 0. The well positions depend on the object’s state. When the object is in state s=0, the lower well is on the left half and the higher well is on the right. Conversely, when the object is in state s=1, the well positions are flipped.

We use the device’s position (*x*) within the area to represent the measurement outcome or, for example, the output symbol. If the device settles in the left half (x=0), it suggests that the input symbol (state) s=0. Conversely, if it ends up in the right half (x=1), it suggests that the input symbol s=1. In essence, the object’s state is transmitted to the device through the potential field, and the device’s final position acts as the received message. The measurement device interprets the potential landscape and settles in the well corresponding to the object’s state. By settling in a specific location (x=0 or x=1), the device receives and reports the information about the object’s state. Based on the above descriptions, the potential profile with respect to the input (*s*) and output (*x*) symbols can be given as follows (see Figure 3):(21)$$\begin{array}{c} U\left(x|s\right)=\left\{\begin{array}{cc} 0, & x=0,s=0\\ H_{0}, & x=1,s=0\\ H_{1}, & x=0,s=1\\ 0, & x=1,s=1 \end{array}\right. \end{array}$$Here, since the potential profiles may not be symmetrical, we employ distinct notations, $H_{0}$ and $H_{1}$, to represent the heights corresponding to different values of *s*. Based on the potential profile in Equation (Equation 21), finding the device in the lower well indicates a precise measurement or correct transmission (x=s). Conversely, if the device resides in the higher well, the measurement outcome or received symbol is imprecise (x≠s).

The device interacts with its environment at a constant temperature (β=1). This environmental noise allows the device to escape local potential minima (wells) and overcome potential barriers within the confined area. As a result, the device reaches equilibrium distributions within the area. These equilibrium distributions describe the probability of finding the device at each location. This is represented by the following equation: (22)$$\begin{array}{c} \left\{\begin{array}{c} p_{l_{0}}=\frac{1}{\exp(-H_{0})+1}\;\mathrm{or}\;p_{l_{1}}=\frac{1}{\exp(-H_{1})+1},\mathrm{probabilities}\;\mathrm{of}\;\mathrm{the}\;\mathrm{device}\;\mathrm{at}\;\mathrm{the}\;\mathrm{lower}\;\mathrm{wells};\\ p_{h_{0}}=1-p_{l_{0}}\;\mathrm{or}\;p_{h_{1}}=1-p_{l_{1}},\mathrm{probabilities}\;\mathrm{of}\;\mathrm{the}\;\mathrm{device}\;\mathrm{at}\;\mathrm{the}\;\mathrm{higher}\;\mathrm{wells}. \end{array}\right. \end{array}$$

Considering the described measurement model, we can interpret the equilibrium distributions (Equation (Equation 22)) in terms of measurement outcomes. The probabilities, $p_{l_{0}}$ and $p_{l_{1}}$, directly reflect the probability of a precise measurement (x=s). Conversely, $p_{h_{0}}$ and $p_{h_{1}}$ represent the probabilities of an imprecise measurement (x≠s). With this understanding of the equilibrium distributions, we can now determine the transmission probabilities q(x|s) as follows: (23)$$\begin{array}{c} q\left(x|s\right)=\left\{\begin{array}{cc} p_{l_{0}}, & x=0,s=0\\ p_{h_{0}}, & x=1,s=0\\ p_{l_{1}}, & x=1,s=1\\ p_{h_{1}}, & x=0,s=1 \end{array}\right. \end{array}$$

During sequential measurements, as depicted in Figure 2, the measurement device remains in a non-equilibrium state as the potential profiles switches based on the state being measured, as shown in Figure 4. In this way, the non-equilibrium strength or the time-irreversible probability flux according to Equation (Equation 7) can be used to represent the degree of non-equilibriumness during the measurement process:(24)$$\begin{array}{c} d=\frac{1}{2}\left[q\left(x=0|s=0\right)-q\left(x=0|s=1\right)\right]=\frac{1}{2}\left(p_{l_{0}}-p_{h_{1}}\right), \end{array}$$
and
(25)$$\begin{array}{c} m=\frac{1}{2}\left(p_{l_{0}}+p_{h_{1}}\right), \end{array}$$
is recognized as the equilibrium strength.

If the states of the object are independently and identically distributed with distribution {P(s=0)=p,P(s=1)=1−p} then the measurement device outputs symbols *x* at t+1 according to distribution (Equation (Equation 1)):(26)$$\begin{array}{c} P\left(x\right)=\left\{\begin{array}{cc} p_{l_{0}}p+p_{h_{1}}\left(1-p\right), & x=0\\ p_{h_{0}}p+p_{l_{1}}\left(1-p\right), & x=1. \end{array}\right. \end{array}$$

The decomposition of the transmission probabilities in this case can be given by Equation (Equation 6) as
(27)$$\begin{array}{c} p_{l_{0}}=m+d,\;\mathrm{and}\;p_{h_{1}}=m-d. \end{array}$$The constraints on *m* and *d* can be given by Equation (Equation 8) as
(28)$$\begin{array}{c} \left\{\begin{array}{c} 0\leq m+d\leq 1\\ 0\leq m-d\leq 1\\ 0\leq m\leq 1 \end{array}\right., \end{array}$$

Then, *d* can be chosen from the convex set D=[a,b] with fixed *m*, where the lower bound a=max(m−1,−m)<0 and the upper bound b=min(1−m,m). More explicitly, D=[−m,m] for 0<m≤1/2, and D=[m−1,1−m] for 1/2<m<1. Here, for numerical illustration, we select four sets of equilibrium strengths *m* and input probabilities P(s=0)=p, which are shown in Table 1.

We show that more non-equilibrium or probability flux leads to more effective information transfer during the measurement. The mutual information between the input and output symbols *s* and *x* as the function of the characterization of the non-equilibrium *d* as the non-equilibrium strength can be derived from Equation (Equation 10):$$\begin{array}{ccc} I & = & S\left(P\left(x=0\right)\right)-pS\left(p_{l_{0}}\right)-\left(1-p\right)S\left(p_{h_{1}}\right)\\ & = & S(m+d(2p-1))-pS(m+d)-(1-p)S(m-d) \end{array}$$
where S(p)=−plogp−(1−p)log(1−p) is the entropy for a binary distribution. In the special case of a symmetric channel, where the potential profiles in Equation (Equation 21) have equal heights ($p_{h}=p_{h_{1}}=p_{h_{0}}$ in Equation (Equation 22)), the information capacity simplifies to Equation (Equation 12):$$\begin{array}{c} C=1-S\left(p_{h}\right)=1-S\left(1/2-d\right). \end{array}$$Here, the optimal input distribution can be given as $P^{*}\left(0\right)=P^{*}\left(1\right)=1/2$; the non-equilibrium strength or the probability flux $d=1/2-p_{h}$ and the equilibrium strength m=1/2 in the symmetric model. We can see from the figure that both the mutual information *I* (Figure 5) and the information capacity *C* (Figure 6a) increase, while the absolute value |d| increases by fixing *m* and the input probability *p*. The minima of *I* and *C* are both achieved at the equilibrium point d=0, and the minima are both zero. This indicates that the transfer efficiency is zero at equilibrium. To see how the information dissipation influences the efficiency of the information transfer, we first evaluate the EPR in Equation (Equation 14) as
$$\begin{array}{c} R=2p(1-p)d[(\log(m+d)-\log(m-d))-(\log(1-m-d)-\log(1-m+d))]. \end{array}$$

By fixing *m* and *p*, *R* has a convex shape with a global minimum at d=0, while it monotonously increases when the absolute value |d| increases (see Figure 7; see also Figure 6b for the information capacity versus the EPR). Therefore, we conclude that in regard to the larger absolute value of the asymmetry of the noise probability |d| or the non-equilibrium strength we characterize the larger non-equilibrium results in both larger mutual information (see Figure 8) and larger information capacity (see Figure 6c) in this model, according to Equations (Equation 19) and (Equation 20), respectively. This leads to larger dissipation cost. In other words, higher mutual information and higher capacity require more dissipation cost to support and sustain.

## 6. Conclusions

This study investigated how non-equilibrium effects influence information transfer efficiency and capacity, quantified by mutual information and information capacity, respectively. We employed the memoryless channel model as a foundation for our exploration. Non-equilibriumness was characterized by the non-equilibrium information driving force, quantified by the probability flux that breaks detailed balance (a key principle in equilibrium systems). Our findings reveal that both mutual information and channel information capacity can be enhanced as the non-equilibrium strength, mediated by the probability flux, increases. This suggests that a stronger non-equilibrium force, from a dynamical perspective, improves both the efficiency and the capacity of information transfer. Additionally, our findings demonstrate that the entropy production rate, which characterizes the thermodynamic cost of dissipation, surprisingly boosts both mutual information and information channel capacity. To illustrate this, we considered a binary channel example. Here, we showed that a more asymmetric noise probability distribution leads to a larger non-equilibrium strength and higher entropy production, ultimately enhancing information transfer efficiency and capacity.

We plan to generalize our model to apply to different physical and biological systems, such as in [32].

## Figures and Tables

**Figure 1 entropy-26-00581-f001:**
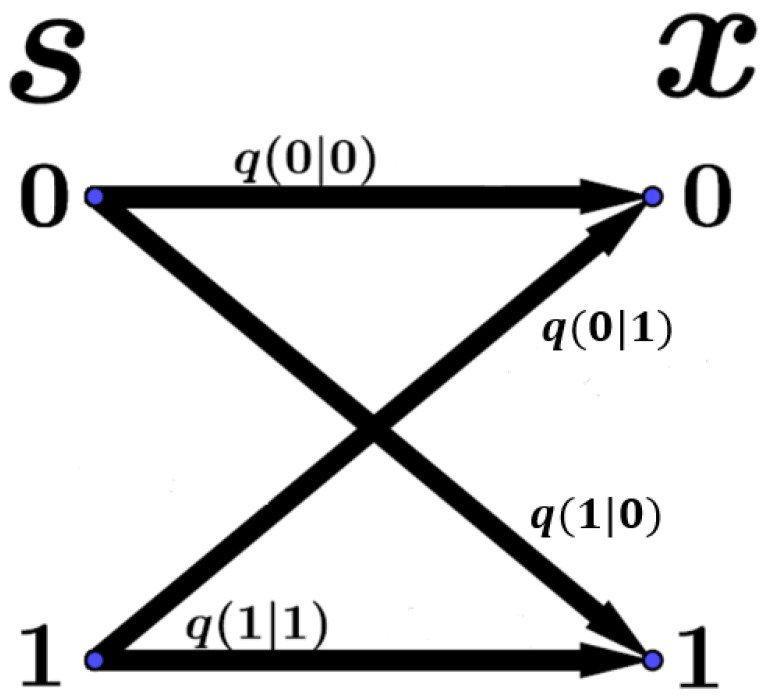
Binary memoryless channel. The input symbol *s* from the information sender is sent to the receiver through the channel. The output symbol is *x*. The channel is characterized by the transmission probabilities q(x|s).

**Figure 2 entropy-26-00581-f002:**
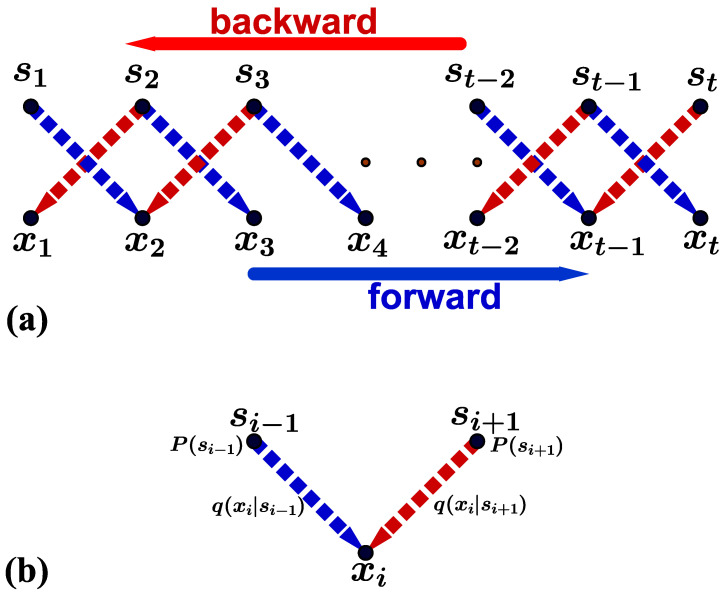
Information processing in the channel: (**a**) A processing time sequence within a time interval *t* is represented by $Z=\{\left(s_{1},x_{1}\right),\left(s_{2},x_{2}\right),...,\left(s_{t},x_{t}\right)\}$. In the forward process (blue arrows), the channel processes the input symbols *s*, and every output symbol $x_{i}$ is determined by the previous symbols $s_{i-1}$ in time, according to the probabilities $q(x_{i}|s_{i-1})$. In the backward process (red arrows), the output symbols $x_{i}$ are determined by the “future” symbols $s_{i+1}$ in time, according to the probabilities $q(x_{i}|s_{i+1})$. (**b**) The minimum graph of the two dynamically correlated transitions $s_{i-1}\to x_{i}$ and $s_{i+1}\to x_{i}$ in the forward and backward time directions, which characterize the time-irreversibility of the dynamics.

**Figure 3 entropy-26-00581-f003:**
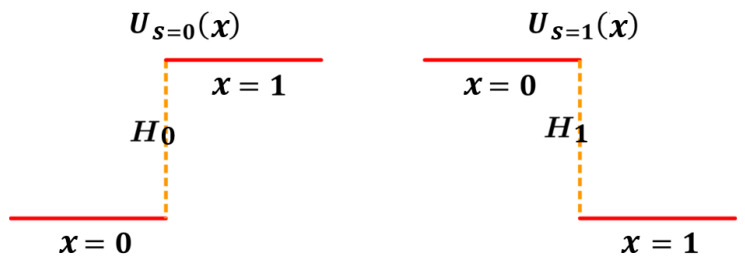
The potential profiles of U(x|s) with two wells described by Equation (Equation 21).

**Figure 4 entropy-26-00581-f004:**
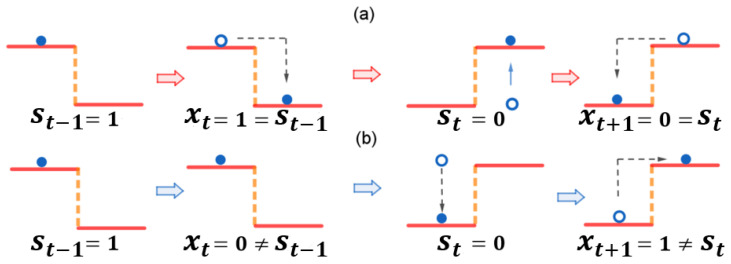
An illustration of the sequential information measurement or transmission. The time sequence of the input symbols is given by $\left\{s_{t-1}=1,s_{t}=0\right\}$. The sequence of the output symbols is given by $\left\{x_{t},x_{t+1}\right\}$: (**a**) the device measures or receives correct messages as $\left\{x_{t}=1,x_{t+1}=0\right\}$, where all the output symbols appear at the correct locations of the lower wells in the potential profiles. (**b**) The device obtains wrong messages as $\left\{x_{t}=0,x_{t+1}=1\right\}$, where all the received symbols appear at the locations of the higher wells in the potential profiles.

**Figure 5 entropy-26-00581-f005:**
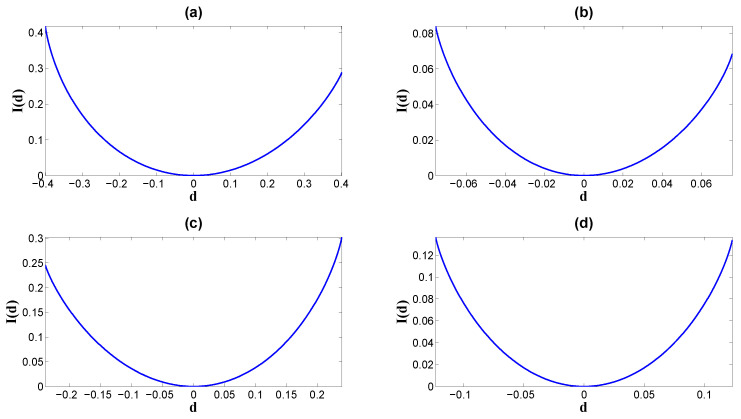
The mutual information rate of the binary channel as the convex function of the non-equilibrium strength *d*, I(d), with fixed equilibrium strength *m* and input probability P(s=0)=p in Table 1: (**a**) Set 1 gives *m* and *p*. (**b**) Set 2 gives *m* and *p*. (**c**) Set 3 gives *m* and *p*. (**d**) Set 4 gives *m* and *p*. The minimum of all the functions I(d) is zero at the equilibrium point where d=0, and I(d) increases monotonously as the absolute value |d| increases.

**Figure 6 entropy-26-00581-f006:**
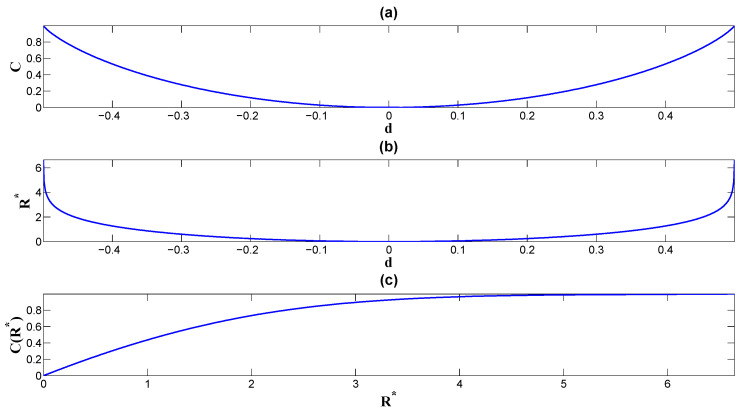
(**a**) The information capacity of the binary channel as the convex function of the non-equilibrium strength *d*, C(d). (**b**) The entropy production rate of the binary channel when the information capacity is achieved; it is the convex function of the non-equilibrium strength *d*, $R^{*}\left(d\right)$. (**c**) The information capacity as the function of the entropy production rate $C(R^{*})$ at optimal input distribution.

**Figure 7 entropy-26-00581-f007:**
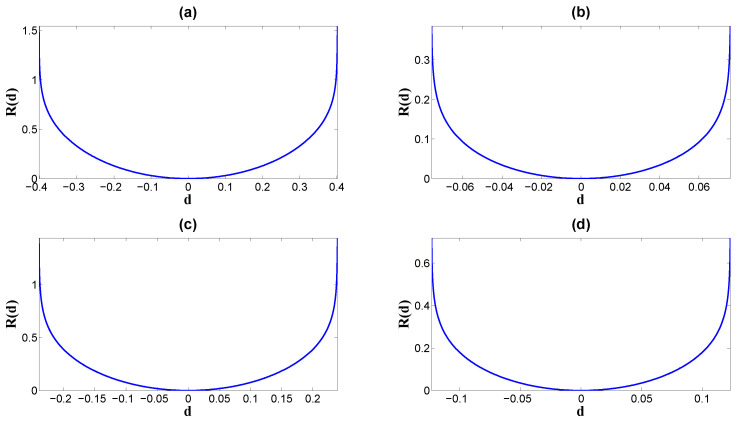
The entropy production rate of the binary channel as the convex function of the non-equilibrium strength *d*, R(d), with fixed equilibrium strength *m* and input probability P(s=0)=p in Table 1: (**a**) Set 1 gives *m* and *p*. (**b**) Set 2 gives *m* and *p*. (**c**) Set 3 gives *m* and *p*. (**d**) Set 4 gives *m* and *p*. The minimum of all the functions R(d) is zero at the equilibrium point where d=0, and R(d) increases monotonously as the absolute value |d| increases.

**Figure 8 entropy-26-00581-f008:**
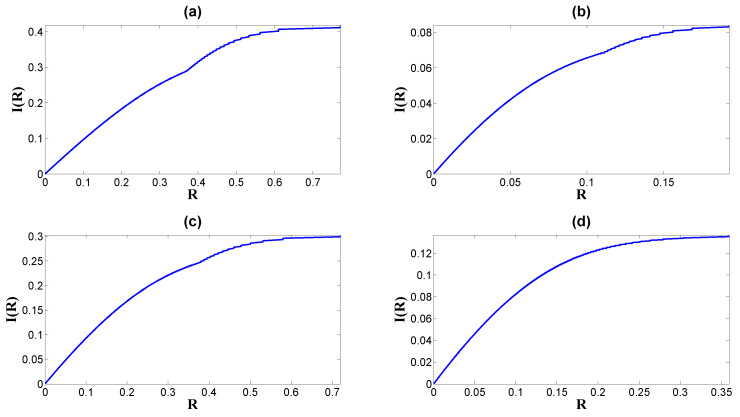
The mutual information as the function of the entropy production rate, I(R): (**a**) The parameters *m* and *p* are given by Set 1 in Table 1. (**b**) Set 2 gives *m* and *p*. (**c**) Set 3 gives *m* and *p*. (**d**) Set 4 gives *m* and *p*.

**Table 1 entropy-26-00581-t001:** Sets of fixed equilibrium strength *m* and input probability P(s=0)=p, used for numerical illustrations.

	Set 1	Set 2	Set 3	Set 4
*m*	0.4018	0.0760	0.2399	0.1233
*p*	0.8530	0.6221	0.3510	0.5132

## Data Availability

The data presented in this study are available on request from the corresponding author.

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
