# Peer review of "Non-Equilibrium Enhancement of Classical Information Transmission"

_entropy, 2024, doi:10.3390/e26070581_

Round 1

Reviewer 1 Report

Comments and Suggestions for Authors

The authors investigated the nonequilibrium effects using memoryless channel model. They found several interesting and new results: 1) the mutual information is a convex function of the nonequilibriumness characterized by the degree of the nonequilibrium information driving force in terms of information flux; 2) the channel information capacity is enhanced by the nonequilibriumness; 3) the nonequilibrium thermodynamic cost characterized by the entropy production rate enhances the mutual information and the information channel capacity.

According to the above results, I will recommend it to be published in Entropy.

The are still a few typos in the manuscript, such as

1)The inequality in line 196: The “greater-than sign” is incorrect.

2)Page 23,Figure 1 is incorrect: the arrow from 1 to 0 is assigned two transmission probabilities q(0|1) and q(1|0). One of them should be assigned to the arrow from 0 to 1.

3)The format in reference is not unified.

Author Response

Dear Reviewer,

We would like to thank you for your positive review and the recommendation for publication in Entropy. We appreciate your time and valuable comments on our manuscript.

We are glad you found our results on the investigation of nonequilibrium effects using the memoryless channel model interesting and novel.  We have carefully addressed your points and revised the manuscript accordingly:

  1. We have confirmed the error in the inequality on line 196. The correct symbol has been incorporated to accurately reflect the convex relationship between mutual information and the degree of nonequilibrium information driving force.
  2. We apologize for the mistake in Figure 1 on page 23. We have corrected the figure by assigning the appropriate transmission probabilities (q(0|1) or q(1|0)) to the corresponding arrows.
  3. We have meticulously reviewed the reference list and ensured a consistent format throughout the manuscript.

We believe these revisions address the identified typos and formatting issues. We have attached a revised version of the manuscript for your reference (highlighting the changes made).

Thank you again for your insightful review. We look forward to your further feedback.

Reviewer 2 Report

Comments and Suggestions for Authors

In this manuscript, the authors investigate the influence of nonequilibrium dynamics on information transfer efficiency and capacity using a memoryless channel model. Nonequilibrium conditions are quantified by the nonequilibrium information driving force, which is associated with information flux breaking detailed balance. The results indicate that increasing the strength of the nonequilibrium information driving force enhances both mutual information and channel capacity. Additionally, the entropy production rate, which relates to the dissipation cost from a thermodynamic perspective, also improves information transfer.

However, to enhance the physical robustness of the findings, I strongly recommend shortening the paper to focus on the essential results and moving additional details into an appendix. This will prevent readers from being overwhelmed by excessive details. Additionally, I have some concerns that I would like the authors to address:

1. Why do you refer to Eq. (5) as information flux? While I understand that this equation represents flux, calling it "information flux" typically requires a "weight" factor that quantifies actual information transfer. For instance, to obtain a charge or energy current, one might multiply the flux by the quantity transferred, such as charge or energy. In the context of information, s (input) and x (output) might be the relevant quantities to consider, necessitating appropriate weighting. The authors should clarify this point.

2. Can the authors quantify the strength of the driving force Q_d in relation to the chosen model parameters? This would assist in understanding the general model of information transfer.

3. Additionally, a graphical visualization of the underlying model would greatly aid in understanding the authors' assumptions.

4. Can authors speculate what might happen if they go from a discrete input-output model to a continuous description like in Entropy 2023, 25(8), 1218; https://doi.org/10.3390/e25081218

In conclusion, I believe this work can be published after the authors significantly shorten the paper for improved readability and address the concerns raised above.

Author Response

Dear Reviewer,

Thank you for your valuable feedback on our manuscript. We have revised the manuscript to incorporate your suggestion. Specifically, we have: 1) Moved detailed derivations and lengthy explanations to an appendix. 2) Retained the main results, core concepts, and essential logical steps in the main body of the paper.

We believe this restructuring will improve the overall readability and accessibility of the manuscript. Readers can now focus on the main ideas and results, while those interested in the finer details can refer to the appendix for additional information.

Next, we will address your thoughtful questions in detail below.

(1). We apologize for any confusion our terminology may have caused. We have revised the manuscript to clarify that the term "information flux" used in the original draft corresponds to the more commonly recognized term "probability flux" in the context of nonequilibrium stochastic dynamics.

The concept of probability flux refers to the time-irreversible flow of probability within a system. In nonequilibrium stochastic dynamics and thermodynamics, this flow arises due to the presence of driving forces and dissipative processes that lead to the system evolving away from equilibrium.

We believe that using the term "probability flux" will align with the standard terminology in the field and enhance the clarity of the manuscript for readers.

(2). Thank you for your insightful comments on our work. We appreciate your emphasis on the importance of clearly highlighting the nonequilibrium aspects of our model.

In response to your feedback, we have further refined our explanation of the nonequilibrium strength, the core concept underpinning the nonequilibrium nature of our information dynamics framework.

As the essence of nonequilibrium in our work, the nonequilibrium strength, represented by the difference between the two transmission probabilities, d = q(x|s) - q(x|s') at the same received symbol x but conditioned on different transmitted symbols s and s’, stands as the fundamental measure of nonequilibrium in our work. This difference quantifies the deviation from equilibrium conditions and serves as the key entity for information transfer.

To emphasize the significance of the nonequilibrium strength, we have placed greater emphasis on its characterization in the revised manuscript. We believe that this prominence aligns with its central role in our nonequilibrium information dynamics framework.

  To further elucidate the physical underpinnings of the nonequilibrium strength, we have incorporated a physical model into the revised manuscript. This model provides a tangible representation of the underlying mechanisms that give rise to the nonequilibrium strength and its role in facilitating information transmission. We detail this case study, which includes: 1) An object existing in two states (s = 0 or s = 1) representing the information source. 2) A measurement device situated within a confined area with a defined potential landscape. 3) The potential landscape featuring two wells with positions dependent on the object's state. 4) The device's position (x) within the area signifying the measurement outcome or received symbol. 5) The equilibrium distribution of the device's position within the potential landscape, influenced by a specific environmental temperature, determines the transmission probabilities q(x|s) for this model. 6) In sequential measurements, the device's equilibrium distribution dynamically evolves due to corresponding switches in the potential profile. This dynamic behavior underlies the emergence of nonequilibrium strength, which quantifies the deviation from equilibrium and drives information transfer in this model.

(3). We appreciate your insightful suggestion. In response, we have incorporated two schematic diagrams (FIG. 3 and FIG. 4) into the new example. These visual aids aim to facilitate readers' comprehension of our ideas from a physical perspective.

(4). You have raised an intriguing question, which is also one of our future research topics. Continuous-state scenarios often involve random walks and Langevin equations. We can utilize the new example in the manuscript to illustrate our understanding of this issue. 1). The information source can be described by Langevin and Fokker-Planck equations. 2). The information receiver perceives a potential field imposed by the source, the shape of which is determined by the source's state (or the message sent by the source). 3). Relative to the rate of change in the source's state, the receiver's state is a fast variable and can quickly reach or approach a steady state in the new potential field formed by the change in the source's state. 4). Based on the above description, the receiver's state or the received message can be regarded as a conditional Markov process conditioned on the message sent by the source, and can be characterized by conditional Langevin and conditional Fokker-Planck equations. 5). Since the external potential field is constantly changing, this information transmission process is a typical non-equilibrium process. We believe that the framework in our work can be used to study this interesting problem. Additionally, we have cited the references you provided in the new version of the manuscript.

Reviewer 3 Report

Comments and Suggestions for Authors

This is a very interesting and rigorously explained essay on the role of nonlinearity (non-equilibrium) in information transmission shedding light on the strict link between irreversibility and information transfer. The Markov formalism is crucial to understand what is going on and, even from an intuitive point of view, the symmetry breaking character of any relevant information transfer is clear. The authors limit themselves to a purely mathematical-theoretical approach referring to further work for real examples. I think the the readers should benefit (even in the present work) from the presentation of some sketchy examples and/or to the reference to some physical counterparts of the model. It is worth noting the correspondence with the stochastic resonance phenomena that were demonstrated to enhance the efficiency of signal recognition in some neural systems (see for example: Hänggi, P. (2002). Stochastic resonance in biology how noise can enhance detection of weak signals and help improve biological information processing. ChemPhysChem3(3), 285-290.).

Author Response

Dear Reviewer,

Thank you again for your insightful and positive review of our manuscript. We appreciate your comments on the importance of the Markov formalism and the clarity of the explanation regarding the link between irreversibility and information transfer.

We particularly valued your suggestion to include a real-world example or reference to a physical counterpart of the model. We believe this addition strengthens the manuscript and improves its accessibility to a broader readership.

Following your recommendation, we have incorporated a new case study that utilizes a simple physical model with binary information measurement. This model serves as a fundamental example of information transmission and allows for clear interpretation of its elements.

In the revised manuscript, we detail this case study, which includes: 1) An object existing in two states (s = 0 or s = 1) representing the information source. 2) A measurement device situated within a confined area with a defined potential landscape. 3) The potential landscape featuring two wells with positions dependent on the object's state. 4) The device's position (x) within the area signifying the measurement outcome or received symbol. 5) The equilibrium distribution of the device's position within the potential landscape, influenced by a specific environmental temperature, determines the transmission probabilities for this model. 6) In sequential measurements, the equilibrium distribution of the device dynamically changes due to corresponding switches in the potential profile. This dynamic behavior reflects the nonequilibrium nature of information measurement and transmission in this model.

We believe this would be a valuable addition to the manuscript and have incorporated a brief discussion of stochastic resonance as a relevant physical example.

Specifically, we have cited the article you recommended:

Hänggi, P. (2002). Stochastic resonance in biology: How noise can enhance detection of weak signals and help improve biological information processing. ChemPhysChem, 3(3), 285-290.

Thank you again for your insightful review. We look forward to your further feedback.

Round 2

Reviewer 2 Report

Comments and Suggestions for Authors

The authors have addressed all the raised concerns and it can be published as is.